# Presence, Tissue Localization, and Gene Expression of the Adiponectin Receptor 1 in Testis and Accessory Glands of Male Rams during the Non-Breeding Season

**DOI:** 10.3390/ani13040601

**Published:** 2023-02-09

**Authors:** Marcelo Martínez-Barbitta, Margherita Maranesi, Francesca Mercati, Daniele Marini, Polina Anipchenko, Luca Grispoldi, Beniamino T. Cenci-Goga, Massimo Zerani, Cecilia Dall’Aglio

**Affiliations:** Department of Veterinary Medicine, University of Perugia, Via San Costanzo 4, 06126 Perugia, Italy

**Keywords:** adiponectin, adiponectin receptors, ovine, ram, testis, sexual glands

## Abstract

**Simple Summary:**

Adiponectin (ADIPOQ) is the most abundant adipocytokine secreted by adipocytes in white adipose tissue and exerts its action by two receptors, ADIPOQ receptor 1 and -2, respectively (ADIPOR1 and -R2). ADIPOQ has an important role in various physiological mechanisms modulating whole-body energy homeostasis. Besides these metabolic aspects, ADIPOQ has been shown to affect the reproductive system through its actions on the hypothalamic–pituitary–gonadal axis. ADIPOQ and its cognate receptors are expressed in different cell types of the male gonad, indicating that this adipocytokine directly regulates the testicular function. To better understand the role of the ADIPOQ/ADIPOQ receptor system in modulating ovine reproductive processes, we have evaluated the ADIPOR1 presence and gene expression in male ram reproductive tissues during the non-breading season. The reported results support the idea that the mammalian reproductive processes are also modulated by the ADIPOQ/ADIPOR1 system, particularly the testicular activity of male rams, during the non-breading season. The study on reproductive activities regulated by the ADIPOQ/ADIPOQ receptors system is helpful for better knowledge of the physiological mechanisms that link adipose tissue with the mammalian reproductive processes, specifically on how altered energy metabolism can induce reproductive pathologies in humans and animals.

**Abstract:**

Adiponectin (ADIPOQ) is a member adipocytokines, and its actions are supported by two receptors, ADIPOQ receptor 1 and -2, respectively (ADIPOR1 and -R2). Our study was performed to evaluate the ADIPOR1 presence and location and its gene expression in reproductive tissues of the male ram, during its non-breading season. The different portions of the male ram reproductive system (testis, epididymis, seminal vesicle, ampoule vas deferens, bulb-urethral gland) were collected in a slaughterhouse. Immunohistochemistry showed ADIPOR1 positive signals in the cytoplasm of all the glandular epithelial cells, with a location near the nucleus; in the testes, the positive reaction was evidenced in the cytoplasm in the basal portion of the germinal epithelial cells. The immune reaction intensity was highest (*p* < 0.001) in the prostate and seminal vesicles glands than that of other parts of the ram reproductive tract. RT-qPCR detected the *ADIPOR1* transcript in the testes, epididymis, vas deferens, bulbourethral glands, seminal vesicles, and prostate; the expression levels were high (*p* < 0.01) in the prostate and low (*p* < 0.01) in the testis, epididymis, and bulbourethral glands. The present results evidenced the possible ADIPOQ/ADIPOR1 system’s role in regulating the testicular activity of male rams during the non-breading season.

## 1. Introduction

Adiponectin (ADIPOQ) is a member of the adipose-secreted proteins, called adipocytokines. The initial report on ADIPOQ in 1995, just one year after the discovery of leptin, was published by Scherer et al. [1]. This molecule is a 244-amino acids protein with a molecular weight of 30 kDa that belongs to the superfamily C1q/TNF-α (tumor necrosis factor-α) [2]. It comprises an N-terminal signal peptide, a collagenous domain, and a globular C1q-like domain at its C-terminus [3]. In sheep, this hormone is encoded by the *ADIPOQ* gene, located on chromosome 1q27 and comprising three exons and two introns [4]. These authors suggested that the *ADIPOQ* gene regulates several productive traits and that sheep with the AA genotype have heavier and larger body dimensions, thereby improving their productivity and reproducibility [4].

ADIPOQ is composed of four distinct domains, which include a signal peptide at the N-terminus, followed by a short variable region, a collagenous domain, and a C-terminal globular domain [5]. ADIPOQ has been found in human and mouse sera as trimeric and hexameric oligomers, although heavy molecular weight forms as well as small proteolytic cleavage products have also been detected [6,7].

ADIPOQ is the most abundant adipose-derived hormone secreted by adipocytes in white adipose tissue, with an important role in the regulation of whole-body energy homeostasis, insulin sensitivity and lipid/carbohydrate metabolism in human and animals [8]. ADIPOQ also plays a role in the stimulation of fatty acid oxidation in the liver and skeletal muscle, suppression of hepatic gluconeogenesis, stimulation of glucose uptake in the skeletal muscle, and increasing insulin secretion [9]. The ADIPOQ actions are supported by two distinct, structurally related, receptors, ADIPOQ receptor 1 and -2, respectively (ADIPOR1 and -R2). These two receptors have been identified in different species, including human [10], rodents [10], chicken [11], pig [12,13], and cow [14].

In addition to its well-known metabolic effects, ADIPOQ has been shown to affect the reproductive system, partially, through central actions on the hypothalamic–pituitary axis [15]. Hypothalamic neurons secrete a gonadotropin-releasing hormone (GnRH) in a pulsatile pattern, stimulating the release of pituitary gonadotropins. These gonadotropins regulate testicular steroidogenesis and spermatogenesis [16]. ADIPOQ receptor R1 and -R2 are generally expressed in the human hypothalamus and pituitary [17], thus suggesting that ADIPOQ could participate in the modulation of the endocrine reproductive axis. ADIPOQ and its cognate receptors are also expressed in different cell types of the male gonad, indicating that this adipocytokine directly regulates the testicular function ADIPOQ through an endocrine and/or paracrine way. In chicken, the presence of the ADIPOQ/ADIPOR1 and -R2 system was evidenced in the seminiferous and peritubular tubule cells [18].

Functional differences and signaling pathways were demonstrated through the generation of ADIPOR1 and -R2 knockout mice: ADIPOR1 related to the activation of AMP-activated and mitogen-activated protein kinase (AMPK) and its pathways [19] and regulates adipose metabolism throughout the regulation of the hormone-sensitive lipase and the peroxisome proliferator-activated receptor (PPAR) γ expression, during adipocyte differentiation [2]. Conversely, ADIPOR2 appears to be associated with the activation of pathways of PPARα [19]. Simultaneous disruption of both ADIPOR1 and -R2 abolished ADIPOQ binding and actions, resulting in increased tissue triglyceride content, inflammation, and oxidative stress, and thus leading to insulin resistance and marked glucose intolerance [19]. Therefore, ADIPOR1 and -R2 serve as the predominant receptors for ADIPOQ in vivo and play important roles in the regulation of glucose and lipid metabolism, inflammation, and oxidative stress in vivo [19].

Recent evidence suggests that ADIPOQ plays a crucial role in mammal reproductive function: ADIPOQ-induced AMPK activation repressed the promoter activity of the kisspeptin1 gene via inhibition of the translocation of specificity protein-1 from the cytoplasm to the nucleus and subsequently influenced GnRH secretion [20]; this AMPK activation by ADIPOQ reduced GnRH-stimulated LH secretion, and this repression was mimicked by 5-aminoimidazole-4-carboxamide riboside, an activator of AMPK [20]

In ovine, the expression of ADIPOQ and ADIPORs has been reported in the male reproductive tract [21] and sperm cells [22]. The latter study also reported that some sperm motility indices (curvilinear velocity, straight-line velocity, average path velocity, linearity, wobble, and straightness) were also significantly correlated with ADIPOQ and ADIPOR1 relative expression, whereas the correlation of ADIPOR2 was also significant with the mentioned parameters, although this correlation was not comparable with ADIPOQ and ADIPOR1 [22]. To better understand the role of the ADIPOQ/ADIPOQ receptor system in modulating ovine reproductive processes, the purpose of this work was to evaluate the ADIPOR1 presence and location and its gene expression in the reproductive tissues of the male ram during the non-breading season.

## 2. Materials and Methods

### 2.1. Collection of Ram Reproductive Tissues

Male reproductive tissues were collected during the non-breading season (May 2021) at Viterbo (Lazio, Italy) slaughterhouse from 12 healthy adult rams (aged 3–8 years, weigh 118–135 kg). The different portions of the reproductive system (testis, epididymis, seminal vesicle, ampoule vas deferens, bulb-urethral gland) of each animal were promptly removed, identified and divided into two fractions, one immediately frozen at −80 °C, and the other fixed by immersion in 4% (*w*/*v*) formaldehyde solution in phosphate buffered solution (PBS) (0.1 M, pH 7.4) for 24 h at room temperature and subsequently processed for embedding in paraffin, following routine tissue preparation procedures.

### 2.2. Immunohistochemistry

The immunohistochemistry method followed that previously reported [23]: 5 µm thick serial sections, mounted on poly-L-lysine coated glass slides using the avidin-biotin complex (ABC, Vector Laboratories, Burlingame, CA, USA) and the chromogen 3,3′diaminobenzidine-4-HCl (DAB, Vector Laboratories). First of all, the sections were dewaxed in xylene and then rehydrated by alcohols in descending percentage. Then, the sections were microwaved three times (5 min at 750 W) in 10 mM citric acid (pH 6.0) for antigen retrieval and cooled at room temperature (15 min). All subsequent steps were performed in a humid chamber at room temperature. Non-specific binding of the primary antibody was prevented by sections’ pre-incubation with the goat normal serum (30 min). The excess liquid was removed, and the sections were incubated (overnight) in the presence of the primary antibody, rabbit polyclonal anti-ADIPOR1 (LS-C151518/55035, 1:100, LSBio, Seattle, WA, USA). The next day, the sections were rinsed in PBS (5 min) and incubated (30 min) with the secondary biotin-conjugated antibody, a goat anti rabbit IgG (BA-1000-1.5, 1:200, Vector Laboratories, Burlingame, CA, USA). Subsequently, they were rinsed (5 min) in PBS and then processed (30 min) with the Vectastain ABC kit (PK-4000, Vector Laboratories) at the manufacturer dilution. The sections were rinsed in PBS, and the reaction was developed with the chromogen solution. After several rinses in PBS, they were counterstained with hematoxylin, dehydrated and mounted in Canada Balsam (BDH, Poole, Dorset, UK). The immunoreaction and the reagents used were validated by positive and negative controls: sections of tissues with the testified presence of the same primary antibody were the positive control [23]; and sections without the presence of the primary antibody and/or replaced with pre-immune mouse-globulin were the negative control of unspecific staining. The intensity of immune reactions was evaluated with the image analysis system (IAAS 2000 image analyzer, Delta Sistemi, Rome, Italy) as described in a previous work [24] through optical density, using five microscope fields of each sample, evaluating the absorbance of the treated tissue in relation to the same without immunohistochemistry treatment.

### 2.3. RNA Extraction and RT-qPCR

Total RNA was purified from the different portions of the genital tract (testis, epi-didymis, seminal vesicle, ampoule vas deferens, bulbourethral gland) of each ram as previously described [25]. Five micrograms of total RNA were reverse transcribed in 20 µL of iSCRIPT cDNA using a random hexamer method according to the protocol provided by the company. Genomic DNA contamination prevention was realized by an RT-qPCR without reverse transcriptase. Serial experiments were carried out to optimize the quantitative reaction, efficiency, and Ct values. In 25 µL RT-qPCR reaction volume were added 12.5 µL of iQ SYBR Green SuperMix (Bio-Rad Laboratories, Hercules, CA, USA), 1 µL forward and 1 µL re-verse primers (stock concentration 10 µM) and 8.5 µL of water. The primers used are listed in Table 1. The final master mix was distributed into a 96-well RT-qPCR plate before adding 2 µL of cDNA for each gene (diluted 10-fold with water). To avoid genomic DNA contamination, for every PCR run, negative reaction controls without reverse transcriptase in RT were performed. Samples’ amplification fidelity was also confirmed by agarose gel electrophoresis. RT-qPCR was carried out in an iCycler iQ (Bio-Rad Laboratories) with an initial incubation at 95 °C for 1.5 min, followed by 40 cycles at 95 °C for 15 s, and 53 °C for 30 s, during which fluorescence data were evaluated. The cycle threshold (CT) value was automatically computed for each trace. The beta-actin Ct housekeeping gene (*ACTB*) was determined to normalize sample variations in the amount of starting cDNA. Standard curves were generated by plotting the Ct against the log cDNA standard dilution (1/5 dilution) in nuclease-free water, and the graph slope was used to determine reaction efficiency. Quantification of the standard curve was evaluated using iCycler system software (Bio-Rad Laboratories), while mRNA gene expression was quantified with the 2^−ΔΔCt^ method [26,27]. The melting curve analysis, performed immediately after the RT-qPCR end cycle, was used to determine the specificity of each primer set. A melt curve protocol was performed by repeating 80 heating cycles for 10 s, from 55 °C with 0.5 °C increments, during which fluorescence data were collected.

### 2.4. Statistical Analysis

Data were analyzed by one-way ANOVA, and multiple comparisons were performed with a Student–Newman–Keuls post hoc t-test. Differences with a probability level of *p* < 0.01 were considered statistically significant. Equality of variances was checked by Levene’s test.

## 3. Results

This is the first publication that reports the histological localization (immunolocalization) of ADIPOR1 in testis and accessory glands of the ram, outside of its reproductive seasonality.

### 3.1. ADIPOR1 Immunolocalization

The immunohistochemical studies revealed a positive signal for ADIPOR1 and evidenced its presence and localization in the cytoplasm of all the glandular epithelial cells. The positive reaction seems to be localized near the nucleus, while the rest of the cytoplasm appeared faintly colored or even negative.

In the testes, the positive reaction was evidenced and localized in the cytoplasm of cells placed in the basal portion of the germinal epithelial cells (arrows) and, also in this case, the localization was peculiarly perinuclear. The particular positivity localization within the cells is typical of many receptors and is an expression of their internationalization after binding to the molecule [29].

The intensity of immune reactions was higher (*p* < 0.001) in the prostate and seminal vesicles glands compared with other parts of the ram reproductive tract (Figure 1 and Figure 2).

### 3.2. Gene Expression

*ADIPOR1* transcripts were detected in the testes, epididymis, vas deferens, bulbourethral glands, seminal vesicles, and prostate (Figure 3). The *ADIPOR1* mRNA expression level was higher (*p* < 0.01) in the prostate and lower (*p* < 0.01) in the testis, epididymis, and bulbourethral glands (Figure 3).

## 4. Discussion

In the present study, we have analyzed the gene and protein expressions and the location of ADIPOR1 in the reproductive tissues of adult male rams. First, our results establish that, for the adult rams, Leydig cells did not express ADIPOR1 differently from other species, where the presence of this receptor and its possible function was found [18,30,31]. Caminos et al. [30], working with rats, found the ADIPOQ IHC presence in adult Leydig cells. Ocón-Grove et al. [18] and Ramachandran et al. [31], also by IHC, established the ADIPOQ presence in chicken Leydig cells. These authors also described the presence of these receptors in Sertoli cells, spermatids, and sperm cells [18,31].

ADIPOR1 was found located in seminiferous tubules of rats [30]. Wu et al. [32] found it in TM3 and mLTC Leydig cell lines in mice, using Western blot, and Ocón-Grove et al. [18] and Ramachandran et al. [31] found it in peritubular locations in chickens.

ADIPOQ and its cognate receptors are also expressed in male reproductive tracts in different species [8,9,14,15,18,21,30]. In particular, they are present in human testes (seminiferous tubules and interstitial tissue), epididymis, Leydig cells and spermatozoa [33]. In mice, the loss of ADIPOR2 induced seminiferous tubular atrophy associated with aspermia and reduction of testes weight [33].

In recent years, Choubey and coworkers [34,35,36,37,38] have extensively studied the role of ADIPOQ on mice testicular activity; in particular, that of the ADIPOQ/ADIPORs system in the prevention of aging and obesity-associated testicular reproductive dysfunctions. In an initial study, these authors [35] reported that ADIPOR1 and -R2 are localized in adult mice Leydig cells and seminiferous tubules. The in vitro study showed the ADIPOQ direct action on spermatogenesis by stimulating cell proliferation (proliferating cell nuclear antigen) and survival and by suppressing cell apoptosis (anti-apoptosis gene Bcl2) [35], thus suggesting an ADIPOQ role in cell survival and proliferation during mice spermatogenesis [35]. Another study [36] reported that, in aged mice testis, the decline in ADIPOQ/ADIPORs system expression is concomitant with that of testicular mass, insulin receptor expression, and testosterone synthesis. In addition, aged mice treated with ADIPOQ showed improvements in testicular mass, cell proliferation, insulin receptor expression, testicular glucose uptake, anti-oxidative enzymes activity and testosterone synthesis [36]. ADIPOQ exogenous administration to type 2 diabetes-induced mice showed an increase of testicular steroidogenic activities, insulin receptor and glucose transporter 8 proteins, and glucose and lactate intra-testicular concentrations [37], thus supporting the idea that ADIPOQ improves testicular functions also through the increase of intra-cellular energy substrate transport and the reduction of oxidative stress [37].

As far as the avians are concerned, in male chickens, the ADIPOQ/ADIPOR1 and -R2 system was expressed in the testes [31]. More precisely, ADIPOQ and ADIPOR1 were localized in the peritubular and Leydig cells, and ADIPOR2 was mainly observed in the Sertoli cells, spermatids, and spermatozoa, suggesting that ADIPOQ can affect the maturation and differentiation of spermatocytes [31].

According to Ocón-Grove et al. [18], in chicken, ADIPOQ and ADIPOR1 immunolocalization and gene expression were evidenced exclusively in the peritubular cells as well as in Leydig cells. Conversely, ADIPOR2 positive cells were found in the ad luminal and luminal compartments of the seminiferous tubules as well as in interstitial cells. In particular, Sertoli cell syncytia, round spermatids, elongating spermatids, spermatozoa, and Leydig cells showed strong ADIPOR2 immunoreactivity.

In agreement with Tabandeh et al. [14], we found that ADIPOR1 immunopositivity was in the cells of the basal portion of germinal epithelium surrounding the seminiferous tubules. Caminos et al. [30] have suggested that the ADIPOQ presence is exclusively in Leydig cells and macrophages in the rat testis interstitium [30]. In addition, these authors [30] reported that *ADIPOR1* mRNA, but not *ADIPOR2*, is present in the seminiferous tubular epithelium isolated from rat testis. In the chicken testis, based on the distinguishable flattened cell morphology of the peritubular cells, *ADIPOQ* and *ADIPOR1* were expressed in peritubular myoid cells [18]. The localization of both *ADIPOQ* and *ADIPOR1* in peritubular cells indicate that ADIPOQ could influence myoid cell function [18]. Peritubular myoid cells are involved in the transport of spermatozoa and testicular fluid from the seminiferous tubule [39], secretion of extracellular matrix proteins such as fibronectin [40], and regulation of Sertoli cell function [39,41]. In addition to myoid cells, the peritubular space also contains immune cells such as macrophages [42]. *ADIPOR1* was shown to be expressed in the epithelium of the seminiferous tubules of rams, where it is involved in the regulation of spermatogenesis, as previously reported in rats [19,30]. Since ADIPOQ has a fundamental role in the male HPG axis and regulation of steroidogenesis [43,44], the effects of circadian disruption on testicular *ADIPOQ, ADIPOR1* and *ADIPOR2* mRNA expressions were examined in some seasonal species [43], finding an inverse relationship between light hours and their gene expression.

Rahmanifar and Tabandeh [21] reported that *ADIPOQ*, *ADIPOR1* and *ADIPOR2* transcripts are present in the testes, epididymitis, and vesicular and bulbourethral glands. Our results demonstrated the location *ADIPOR1* in cells of glandular epithelium in adult rams. Additionally, the *ADIPOR2* expression level in different parts of the male reproductive tract was more than that of *ADIPOR1* [21]. Unfortunately, this study lacks the date of sampling, and these data would have been important to understand the possible seasonality of the ADIPOQ/cognate receptors system. In mammals, testicular growth and regression are photoperiod-dependent, meaning they are mainly determined by the endogenous circadian secretion of melatonin [45]. In mammals, endogenous biological rhythms regulate multiple physiological and behavioral processes that are essential for successful reproduction, so much that their misalignment provokes reproductive disorders [45]. Within this context, in rats, a species that shows a circadian rhythm, Moustafa [46] demonstrated that *ADIPOR1* is expressed in the epithelium of the seminiferous tubules, where it is involved in the regulation of spermatogenesis [30]. Since ADIPOQ has a fundamental role in the male HPG axis and regulation of steroidogenesis, the effects of circadian disruption on testicular *ADIPOQ*, *ADIPOR1* mRNA expressions in the ram should be examined, as has been demonstrated in other species.

Ovine are known as mammals with a marked seasonality of breeding activity. In fact, in these species, the daily photoperiod is the determinant factor for this activity, whereas environmental temperature, nutritional status, and social interactions are modulators, as evidenced by the correlation between AMPK and variable gonadotropins [19] during the ovine cycle. Taibi et al. [47] indicated that AMPK is expressed in the ovine testis and regulates steroidogenesis in male sheep. Additionally, pituitary AMPK is known to act as an energy sensor, thus controlling gonadotropin secretion and reproduction in bovine [20]. In this context, it is important to emphasize that AMPK is activated by ADIPOQ to inhibit human GnRH release, through the hyperpolarization of plasma membranes as well as calcium influx [48]. In addition, Dutta et al. [49] reported that the amount of GnRH immunoreactive neurons decreased with ADIPOQ mutations, suggesting that this cytokine controls GnRH secretion in mammal hypothalamus. The important role of ADIPOQ in reproductive mechanisms is also suggested by its effects on prostaglandins secretion [50]. In particular, it is well known that the semen of mammalian species contains high amounts of different prostaglandins. These findings support the idea that the ADIPOQ/cognate receptor system may be associated with the secretion of these factors in the male reproductive tract.

## 5. Conclusions

The present results strengthen the evidence of the ADIPOQ/ADIPOR1 system’s role in regulating the mammalian reproductive processes, particularly in the testicular activity of male rams, during the non-breading season. Despite this, our knowledge is still underdeveloped; therefore, future studies are needed to better elucidate the fine mechanisms of the ADIPOQ/ADIPOQ cognate receptors system in modulating reproductive processes. This future research on reproductive activities regulated by the ADIPOQ/ADIPOQ receptors system will enable us to better understand the physiological mechanisms that link adipose tissue with the mammalian reproductive processes, specifically on how an altered energy status can induce reproductive pathologies in humans and animals.

## Figures and Tables

**Figure 1 animals-13-00601-f001:**
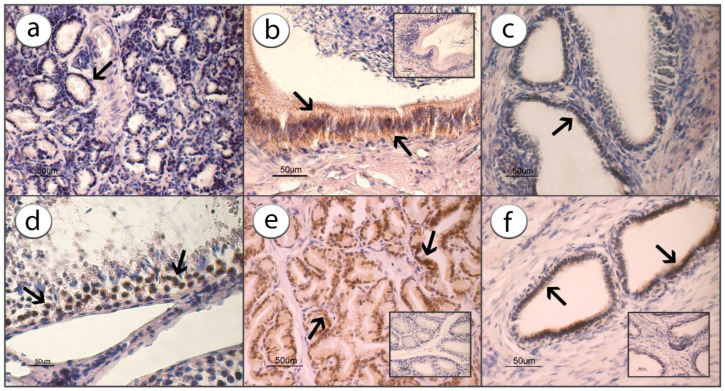
Immunostaining for ADIPOR1 in bulbourethral gland (**a**), epididymis (**b**), seminal vesicle (**c**), testicle (**d**), prostate (**e**) and vas deferens (**f**) counterstained with hematoxylin. The arrows indicate the positive localization of the immunoreaction, while the inserts in (**b**,**e**,**f**) are examples of the negative reactions.

**Figure 2 animals-13-00601-f002:**
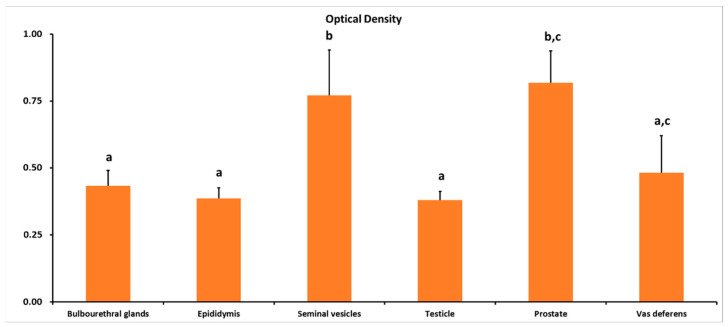
Immunoreaction intensity of ADIPOR1, performed on adult ram male reproductive tissues. Different letters above bars indicate significantly different values (ANOVA *p* < 0.001, Levene’s test *p* > 0.05).

**Figure 3 animals-13-00601-f003:**
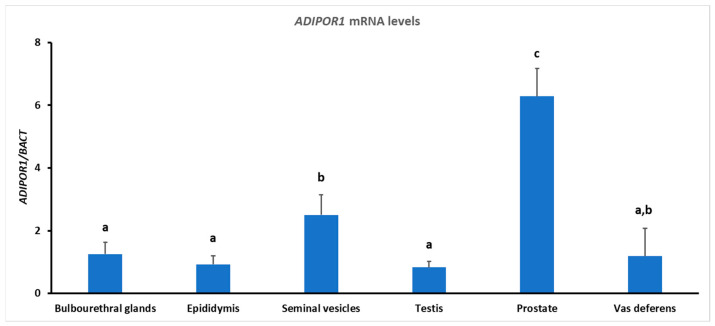
RT-qPCR analysis of *ADIPOR1* gene expressions performed on adult ram male reproductive tissues. Different letters above bars indicate significantly different values (ANOVA *p* < 0.001, Levene’s test *p* > 0.05).

**Table 1 animals-13-00601-t001:** Primers for *ADIPOR1* and *ACTB* [28] housekeeping gene used for RT-qPCR quantification.

Gene	NCBI Seq. Ref.		Primers	Bp
*ADIPOR1*	NM_001306110.1	F	GGTGGTGTTCGGGATGTTCT	128
		R	CGATCCCCGAATAGTCCAGC	
*ACTB*	U39357.2	F	CCTTAGCAACCATGCTGTGA	130
		R	AAGCTGGTGCAGGTAGAGGA	

## Data Availability

The data presented in this study are available on request from the first author/corresponding authors.

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
