# Peer review of "Presence, Tissue Localization, and Gene Expression of the Adiponectin Receptor 1 in Testis and Accessory Glands of Male Rams during the Non-Breeding Season"

_animals, 2023, doi:10.3390/ani13040601_

Round 1

Reviewer 1 Report

The main goal of the study was determine the presence, location and expression of Adiponectin receptor 1 (ADIPOQR1) in the ovine male reproductive tract during the non-breeding season. The authors were able to detect the presence of the receptor in testis, epididydimis and accesory sex glands using immunohistochemistry and RT-qPCR. The authors have found that prostate and vesicle glands have the higest transcript and protein expression than that observed in other parts of the male tissues. One of the main finding was that protein expression of the receptor was found in the cytplasm of only germinal cells of the semminiferous tubules ulike the location in leydig cells reported in several species by different authors-

The manuscript is very well written and easy to follow, with a solid background that justify the main goal of the study.

Minor concerns

-Revise the sentence on line 17 Page 1, where is written both............

-Revise the sentence on line 37 Page 1, was highest in prostate and vesicle than that of...................................

-Transcrip line 38 page 1

-See sentence: was high and low, line 40

-Check the paragraph, line 213 to 217

-line 217, the authors also described

-line 220, using western blot, WT??, see the sentence

Author Response

Reviewer 1

-Revise the sentence on line 17 Page 1, where is written both

Sentence revised, see line 17

-Revise the sentence on line 37 Page 1, was highest in prostate and vesicle than that of

Sentence revised, see lines 37-38 17

-Transcrip line 38 page 1

Error corrected, see line 38

-See sentence was high and low, line 40

Sentence revised, see line 40

-Check the paragraph, line 213 to 217

Paragraph revised, see lines 218-221

-line 217, the authors also described

Sentence revised, see line 223

-line 220, using western blot, WT??, see the sentence

Sentence revised, see line 226

Reviewer 2 Report

The Martínez-Barbitta et al., 2022, manuscript ID 2182483 addresses the immunolocalization and gene expression of the adiponectin receptor 1 in the testis and other accessory sex glands of male rams during the non-breeding season.

There are few queries and few suggestions which makes this manuscript more representable to be publish.

1.       The authors need to include a proper labeling of histological slide of reproductive organs in mice?

2.       The authors fails to cite the recent articles (PMID: 30471430, 29908833) published on immunological localization of adipoR1 in the adult, aging, diabetic testis. Further the authors cite the article but the paragraph mentioned belongs to other article of same author (29908833 replaced by 30471430). It need to be corrected the reference (ref 31) in discussion section. You have to include all four citations.

3.       Why the authors have not tried the adipoR2 receptor and adipoq immunostaining in the similar groups of organs. That will be more informative to see there expression in non-breeding season rams.

4.       Do the authors check the status of insulin resistance in these rats?

5.       Is it possible to see the protein expression of these receptors or levels of adiponectin in breeding and non-breeding animals?

Author Response

Reviewer 2

1. The authors need to include a proper labeling of histological slide of reproductive organs in mice?

We do not understand this request, each photo shown in the figure has a letter superscribed, to which the testicular tissue corresponds in the legend.

2. The authors fails to cite the recent articles (PMID: 30471430, 29908833) published on immunological localization of adipoR1 in the adult, aging, diabetic testis. Further the authors cite the article but the paragraph mentioned belongs to other article of same author (29908833 replaced by 30471430). It need to be corrected the reference (ref 31) in discussion section. You have to include all four citations.

Citations added, see references 34-38. The discussion has been expanded in relation to the added citations, see lines 233-250

3. Why the authors have not tried the adipoR2 receptor and adipoq immunostaining in the similar groups of organs. That will be more informative to see there expression in non-breeding season rams.

We are working on this, unfortunately technical problems, antibodies for ADIPOQ and ADIPOR2 do not give results, even on control tissues

4. Do the authors check the status of insulin resistance in these rats?

The rams come from the slaughterhouse, so we know their general health status but not other particular clinical conditions.

Round 2

Reviewer 2 Report

I hope to see future work related to Adipoq and adipoR2 antibodies. The authors have tried to justify the comments raised by me.